# Incidence of tuberculosis among PLHIV on antiretroviral therapy who initiated isoniazid preventive therapy: A multi-center retrospective cohort study

Andrew Kazibwe[1,2], Bonniface Oryokot[1,3]*, Levicatus Mugenyi[1], David Kagimu[1], Abraham Ignatius Oluka[1], Darlius Kato[1], Simple Ouma[1], Edmund Tayebwakushaba[1], Charles Odoi[1], Kizito Kakumba[1], Ronald Opito[1], Ceasar Godfrey Mafabi[1], Michael Ochwo[1], Robert Nkabala[1], Wilber Tusiimire[1], Agnes Kateeba Tusiime[1], Sarah Barbara Alinga[1], Yunus Miya[1], Michael Bernard Etukoit[1], Irene Andia Biraro[2], Bruce Kirenga[2,4]

**1** The AIDS Support Organization (TASO), Kampala, Uganda, **2** Makerere University School of Medicine, Kampala, Uganda, **3** University of Suffolk, Ipswich, United Kingdom, **4** Makerere University Lung Institute, Kampala, Uganda

* bonory@gmail.com

**Data Availability Statement:** Minimally anonymized data are within the Supporting Information files.

## Abstract

### Introduction

Isoniazid preventive therapy (IPT) is effective in treating tuberculosis (TB) infection and hence limiting progression to active disease. However, the durability of protection, associated factors and cost-effectiveness of IPT remain uncertain in low-and-middle income countries, Uganda inclusive. The Uganda Ministry of health recommends a single standard-dose IPT course for eligible people living with HIV (PLHIV). In this study we determined the incidence, associated factors and median time to TB diagnosis among PLHIV on Antiretroviral therapy (ART) who initiated IPT.

### Materials and methods

We conducted a retrospective cohort study at eleven The AIDS Support Organization (TASO) centers in Uganda. We reviewed medical records of 2634 PLHIV on ART who initiated IPT from 1st January 2016 to 30th June 2018, with 30th June 2021 as end of follow up date. We analyzed study data using STATA v.16. Incidence rate was computed as the number of new TB cases divided by the total person months. A Frailty model was used to determine factors associated with TB incidence.

### Results

The 2634 individuals were observed for 116,360.7 person months. IPT completion rate was 92.8%. Cumulative proportion of patients who developed TB in this cohort was 0.83% (22/2634), an incidence rate of 18.9 per 100,000 person months. The median time to TB diagnosis was 18.5 months (minimum– 0.47; maximum– 47.3, IQR: 10.1–32.4). World Health Organization (WHO) HIV clinical stage III (adjusted hazard ratio (aHR) 95%CI: 3.66 (1.08,

**Funding:** The authors received no specific funding for this work.

**Competing interests:** The authors have declared that no competing interests exist.

12.42) (P = 0.037) and discontinuing IPT (aHR 95%CI: 25.96(4.12, 169.48) (p = 0.001)), were associated with higher odds of TB diagnosis compared with WHO clinical stage II and IPT completion respectively.

## Conclusion

Incidence rates of TB were low overtime after one course of IPT, and this was mainly attributed to high completion rates.

## Introduction

The World Health Organization (WHO) estimates that 1.7 billion individuals globally harbor tuberculosis (TB) infection (TBI) and approximately 5–10% may progress to TB disease in their lifetime [1, 2]. In 2020, nearly ten million individuals developed illness due to TB with 1.3 million TB-related deaths recorded globally [3]. During the same period, people living with HIV (PLHIV) accounted for 792,000 and 214,000 of TB cases and deaths respectively. Disproportionately, TB morbidity and mortality among PLHIV is higher in TB/HIV high-burden countries like Uganda [4]. PLHIV possess a 5–10% annual risk of developing active TB disease compared to their HIV negative counterparts whose lifetime risk is only about 10–20% [5–7]. Moreover, PLHIV have higher risk of poor TB treatment outcomes compared to HIV negative individuals [8]. Furthermore, one in every three AIDS-related deaths is attributable to TB and its complications [9]. Therefore, prevention of TB, through treatment of TBI confers significant morbidity and mortality benefits among PLHIV.

Isoniazid preventive therapy (IPT), when combined with ART, offers between 60–90% protection against progression of TBI to active tuberculosis disease [2, 10–13]. As such, provision of IPT, particularly among the PLHIV, is a global health priority as enshrined in the Global HIV/AIDS Strategy 2021–2026 [14], WHO End TB Strategy [15] and the President's Emergency Plan for AIDS Relief (PEPFAR) Country Operational Plan (COP) 21 [16]. This position was unanimously emphasized at the United Nations High level meeting on tuberculosis in 2018 during which countries committed to providing TB preventive therapy to six million PLHIV by 2022 [2, 6, 17, 18]. Concerns however, remain over the persistently high incidence, morbidity and mortality due to TB among PLHIV in Uganda [19, 20]. This has been attributed to low completion of IPT and other patient factors; based on which Aashna et al. consequently argue that IPT may be less cost-effective in low and middle income compared to higher income countries [12]. In addition, uncertainty of duration of IPT protectiveness among some PLHIV is also of concern [9, 10, 21, 22].

Whereas introduction of new, shorter duration IPT regimens is expected to address non-completion, the duration of protection remains contentious [23]. For example, low CD4 count of less than 200 cells, WHO clinical stage 3 or 4, male sex, unemployment and underweight appear to have significant association with active TB disease development, after IPT completion [21, 22]. Thus, the WHO guidelines on programmatic management for TBI recommend repeat or longer duration of IPT in high-risk PLHIV such as contacts of confirmed TB patients [11]. However, the Uganda MOH still implements a largely "one-size-fits-all" IPT regimen for PLHIV, and this is probably due to the paucity of data on incident TB cases among PLHIV who initiate IPT in Uganda [24]. We present the results of a multi-center retrospective cohort study that measured the incidence, associated factors and median time to TB diagnosis among PLHIV who had received IPT.

## Materials and methods

### Study design

This was a retrospective cohort study, involving analysis of routinely collected clinical data. At the end of 30[th] June 2021, we evaluated data of PLHIV who were initiated on IPT from 1[st] January 2016 to 30[th] June 2018 and followed up for a minimum of 36 months.

### Study setting

The study was conducted at The AIDS Support Organization (TASO) Uganda Centers of Excellence (COEs). Mr. and Mrs. Kaleeba led a team of sixteen volunteers in 1987 to found TASO Uganda to contribute to a process of HIV prevention, restore hope and improve the quality of life of individuals, families and communities affected by the pandemic [25, 26]. Through the 11 COEs, the organization provided comprehensive HIV/TB services to 78,897 PLHIV by the end of March 2021. It is noteworthy that TASO Uganda, through its COEs, started providing IPT to PLHIV in 2015 with particular focus on children aged under 15 years. IPT was integrated in HIV clinical care based on the 1998 WHO and Uganda Ministry of Health (MOH) guidelines for a 6–9 months' course of Isoniazid (INH) preferred [22, 27]. Sub-optimal uptake of IPT prompted MOH with partners to implement an ambitious scale up campaign dubbed '100 days of IPT' aimed at scaling up IPT enrolment of PLHIV. Through this intervention, Uganda enrolled over 300,000 PLHIV on IPT in July-September 2019 quarter [28]. PLHIV were screened clinically to rule out active TB disease using the WHO intensified case finding (ICF) form [29] before initiating IPT. PLHIV at TASO COEs received a 5-10mg/kg or a maximum of 300mg/day daily dose of isoniazid, co-administered with 25-50mg/day of pyridoxine for prevention of peripheral neuropathy. The PLHIV who received IPT also benefitted from routine adherence counselling, and regular monitoring for drug side effects and repeat screening for active TB.

### Study population

All PLHIV who received IPT from 1[st] January 2016 to 30[th] June 2018 at TASO COEs were eligible for inclusion in the study. Individuals with incomplete data on IPT enrolment or completion dates, were excluded from the study.

### Data extraction

We abstracted a list of 2634 eligible participants from the Uganda Electronic Medical Records and TASO Management Information System at each TASO COE. This enabled identification and retrieval of individual patient file for physical verification and extraction of the required data for entry. Data were directly entered into an online-based questionnaire powered by KoboToolbox® [30]. Where certain variables were found missing in patient files, patient registers including facility-based TB registers, presumptive TB registers, ART registers, viral load and IPT registers were used.

### Study variables

We collected minimally anonymized patient specific data on demographic and baseline clinical characteristics. Key variables included Age, Gender, baseline WHO clinical stage, IPT enrolment and outcome dates, Viral load status, ART regimens, ART status, baseline CD4 count and baseline functional assessment.

## Exposure and outcome variables

The primary exposure was IPT initiation. We considered PLHIV on ART, who were initiated on daily INH from 1st/January/2016 to 30th/June/2018 and retrospectively followed them up for at least three years or until censorship. Those who were already on IPT were excluded from the study. We also abstracted data on IPT outcome as defined by MOH: completed–chart documentation of patient self-report and dispensing record of completion of standard IPT course; stopped–discontinued IPT because of drug stock-outs, side effects, clinician's decision to interrupt IPT (in case of suspected drug-drug interactions, contraindications) and diagnosis of TB. Patients were documented as LTFU if they could not be returned to care after interruption, for at least three months despite active follow up. Patients were reported as dead if patient's next of kin reported that patient had died and this verified by the counsellor through a home visit or local leader report. Information on causes of death was not available for this study.

The primary outcome was TB disease diagnosis. We stratified this by, pulmonary bacteriologically confirmed or clinically diagnosed disease, or extra-pulmonary TB. We utilized the MOH diagnostic algorithm to determine date of TB diagnosis; the date when the attending clinician made a decision to treat the patient for TB, the date of a positive microbiological test, or the date of entry into the unit TB register, whichever occurred earlier. In addition, secondary outcome of interest included time to TB diagnosis. We measured time to TB diagnosis from the date of patient enrolment on IPT to the date of TB diagnosis.

## Censorship

All patients who were diagnosed with TB during the follow-up period were censored on the date of TB diagnosis. Those who did not develop TB, but died before completing the follow up period were censored on the reported date of death, while those who were still alive were censored at the pre-set end of the study period, 30th June 2021. This date allowed for a minimum follow up period of three years (36 months). Finally, for patients who were transferred out, lost to follow up and those who missed appointment, observation was censored on the last clinical encounter date.

## Statistics and data analysis

Study data were downloaded as Microsoft Excel spreadsheets, cleaned and exported to STATA v.16.0 (STATA Corp, College Station, TX) for analysis. Study variables were summarized using absolute numbers and proportions. Continuous exposure variables such as age, baseline CD4, duration on ART were converted to categorical variables. Cumulative TB incidence was computed as the number of new TB cases expressed as a percentage of the total number of individuals at risk. Incidence rate of TB was computed as the number of new TB cases divided by the total person time in months presented as per 100,000 person months. To determine factors associated with incidence of TB while accounting for facility-level clustering, a Frailty model was used. Hazard ratio (HR) with 95% CI was used to compare the risk for TB diagnosis between factor-levels. Factors with p-value less than 0.2 at bivariable analysis were all subjected to a multivariable analysis and results from the latter presented as adjusted estimates. A p-value <0.05 was considered statistically significant.

## Ethical approval

We received ethical approval from TASO Research Ethics Committee (REC) (TASOREC/050/2021-UG-REC-009). Since our study only involved secondary data without direct engagement of study participants, REC exempted us from obtaining informed consent from individual patients.

## Results

### Patient demographics

A total of 3688 individuals were initiated on IPT during the study period but 1034 were excluded due to incomplete information (see Table 1 attached as supporting document). Therefore, we collected data on the remaining 2634 PLHIV. Of the 2634 PLHIV on ART who were initiated on IPT, majority were females (65.6%), had been on ART for more than 36 months (63.3%) and initiated IPT in 2017 (67.1%) (Table 1). Nearly a third of the participants were children aged less than 15 years (32.6%). At ART initiation, 99.8% of them had a functional status of Working/Playing, 30.6% did not have baseline CD4 done, 46.7% were initiated on a Tenofovir-backbone regimen and 95.9% were on a non-nucleoside reverse transcriptase anchor regimen. Prior to IPT initiation, 32.1% had not had a viral load done and 41.2% had a detectable viral load (Table 2).

In this study, 92.8% of the participants were documented to have completed the IPT course. However, among the 24 who discontinued, the reasons for discontinuation included drugs out of stock 6 (25%); side effects 5 (21%), treatment interruption 5 (21%), while 6(25%) were diagnosed with active TB while still on IPT and had to initiate the full course of anti-TB treatment while 2 (8%) transferred out of care.

A total of 22 participants were diagnosed with TB during this study, majority (8/22, 36.4%) of whom were diagnosed in 2019. For the other years, the numbers were 3 (13.6%) in 2017, 5 (22.7%) in 2018, 4 (18.2%) in 2020, and 2 (9.1%) in 2021. Of those diagnosed with TB, 8 (36.4%) had pulmonary bacteriologically confirmed (PBC) disease, 6 (27.3%) were pulmonary clinically diagnosed (PCD) disease while 8 (36.4%) had extrapulmonary tuberculosis disease.

### Incidence of TB and associated factors

The cumulative incidence of TB was 0.83% (22/2634). Among those diagnosed with TB, 14 (63.6%) were females, 11 (50.0%) were children less than 15 years of age, 77.3% had baseline WHO clinical stage of II, 31.8% had a baseline CD4 greater than 500 cells/ml, 15 (68.2%) had been on ART for more than 36 months, 13 (59.1%) were on an AZT backbone regimen, 12 (54.5%) were on an Efavirenz-based anchor regimen, 18 (81.8%) had completed IPT. The median time from IPT initiation to TB diagnosis was 18.5 months (interquartile range, IQR: 10–32) months (Table 3).

**Table 1. Number of PLHIV enrolled on IPT from 1st January 2016 to 30th June 2018, and number enrolled into the study.**

| TASO COE | PLHIV enrolled on IPT from 1st/January/2016 to 30th/June/2018. | PLHIV enrolled into the study (%) |
|---|---|---|
| Entebbe | 670 | 37 (5.5) |
| Gulu | 381 | 378 (99.2) |
| Jinja | 647 | 430 (68.5) |
| Masaka | 295 | 283 (95.9) |
| Masindi | 562 | 484 (86.1) |
| Mbale | 231 | 224 (97.0) |
| Mbarara | 62 | 61 (98) |
| Mulago | 162 | 100 (61.7) |
| Rukungiri | 48 | 44 (102.1) |
| Soroti | 128 | 118 (93.8) |
| Tororo | 482 | 475 (98.5) |
| **Total** | **3668** | **2634 (72%)** |

**Table 2. Distribution of demographic and clinical characteristics of the study participants.**

| Characteristics | Total (N = 2634) (%) | Characteristics | Total (N = 2634) (%) |
|---|---|---|---|
| **Sex** | | **Baseline ART regimen backbone** | |
| Female | 1728 (65.6) | | |
| Male | 906 (34.4) | ABC-based regimen | 343 (13.0) |
| **Age** | | AZT-based regimen | 992 (37.7) |
| <15 years | 859 (32.6) | D4T-based regimen | 70 (2.7) |
| 15–24 years | 222 (8.4) | TDF-based regimen | 1229 (46.7) |
| 25–34 years | 465 (17.7) | **Baseline ART regimen anchor** | |
| 35–44 years | 542 (20.6) | DTG | 81 (2.6) |
| 45–54 years | 375 (14.2) | PI | 43 (1.4) |
| 55+ years | 171 (6.5) | NNRTI | 2998 (95.9) |
| **Marital status** | | Triple NRTI | 1 (0.0) |
| Child | 779 (29.6) | **Viral Load status prior to IPT initiation** | |
| Married | 1035 (39.3) | Detectable | 1085 (41.2) |
| Separated/Divorced | 294 (11.2) | Undetectable | 704 (26.7) |
| Single | 355 (13.5) | Not Done | 845 (32.1) |
| Widowed | 171 (6.5) | **Year of IPT initiation** | |
| **Baseline WHO clinical stage** | | 2016 | 256 (9.7) |
| I | 176 (6.6) | 2017 | 1768 (67.1) |
| II | 2235 (84.9) | 2018 | 610 (23.2) |
| III | 182 (6.9) | **IPT outcome** | |
| IV | 41 (1.6) | Completed | 2444 (92.8) |
| **Baseline CD4 count** | | LTFU | 162 (6.2) |
| <200 cells/ml | 595 (22.6) | Stopped | 24 (0.9) |
| 200–350 cells/ml | 470 (17.8) | Died | 4 (0.2) |
| 350–500 cells/ml | 285 (10.8) | | |
| Above 500 cells/ml | 478 (18.2) | **Baseline functional status** | |
| Not Done | 806 (30.6) | Working/Playing | 2628 (99.8) |
| **Duration on ART at IPT initiation (months)** | | Ambulatory | 4 (0.2) |
| <6 months | 250 (9.5) | Bed-ridden | 2 (0.1) |
| 6-<12 months | 175 (6.6) | | |
| 12-<18months | 103 (3.9) | | |
| 18-<24 months | 121 (4.6) | | |
| 24 - <30 months | 122 (4.6) | | |
| 30 - <36 months | 196 (7.4) | | |
| 36+ months | 1667 (63.3) | | |

Table 3 shows the number diagnosed with TB, total person months, incidence rate and Hazard Ratio (HR) by patients' demographic and clinical characteristics. Overall, the incidence of TB per 100,000 person months was 18.9 (95% CI: 12.4, 28.7) and it was higher among males than females (Crude hazard ratio (cHR) (95% CI): 1.17 (0.49, 2.80), p = 0.724). The incidence of TB was significantly lower among patients in the age group 25–34 years compared to those aged below 15 years (cHR (95% CI): 0.11 (0.01, 0.88), p = 0.037) but not between the other age groups. Patients in the WHO clinical stage III had higher incidence of TB compared to their counterparts in stage II (cHR (95% CI): 4.27 (1.28, 14.29), p = 0.018). The incidence was lower among patients on baseline TDF-based ART regimen compared to those on AZT (cHR (95% CI): 0.27 (0.10, 0.79), p = 0.016). Patients who stopped IPT had a much higher hazard for TB infection compared to those who completed (cHR (95% CI): 14.54 (3.05, 69.13), p = 0.001).

**Table 3. Incidence of TB and associated factors.**

| | TB diagnosis n (%) | Total person months | Crude incidence rate per 100,000 person months (95% CI) | Crude HR* (95% CI) | p-value | Adjusted HR* (95% CI) | p-value |
|---|---|---|---|---|---|---|---|
| **Overall** | **22** | **116360.7** | **18.9 (12.4, 28.7)** | | | | |
| **Gender** | | | | | | | |
| Female | 14 (63.6) | 76132.1 | 18.4 (10.9, 31.0) | Ref. | | | |
| Male | 8 (36.4) | 40228.6 | 19.9 (9.9, 39.8) | 1.17 (0.49, 2.80) | 0.724 | | |
| **Age** | | | | | | | |
| <15 years | 11 (50.0) | 39484.4 | 27.9 (15.4, 50.3) | Ref. | | Ref. | |
| 15–24 years | 1 (4.5) | 9671.8 | 10.3 (1.5, 73.4) | 0.25 (0.03, 1.97) | 0.188 | 0.29 (0.04, 2.35) | 0.246 |
| 25–34 years | 1 (4.5) | 20094.8 | 5.0 (0.7, 35.3) | 0.11 (0.01, 0.88) | 0.037 | 0.18 (0.02, 1.68) | 0.132 |
| 35–44 years | 3 (13.6) | 23314.1 | 12.9 (4.2, 39.9) | 0.32 (0.09, 1.20) | 0.092 | 0.36 (0.08, 1.59) | 0.178 |
| 45–54 years | 4 (18.2) | 16405.9 | 24.4 (9.2, 65.0) | 0.66 (0.20, 2.17) | 0.495 | 0.73 (0.19, 2.74) | 0.636 |
| 55+ years | 2 (9.1) | 7389.6 | 27.1 (6.8, 108.2) | 0.76 (0.16, 3.52) | 0.723 | 0.89 (0.18, 4.40) | 0.886 |
| **Baseline WHO clinical stage** | | | | | | | |
| I | 0 (0.0) | 7906.1 | 0.0 | 0.0 | - - - | 0.0 | |
| II | 17 (77.3) | 98405.0 | 17.3 (10.7, 27.8) | Ref. | | Ref. | |
| III | 4 (18.2) | 8251.4 | 48.5 (18.2, 129.2) | 4.27 (1.28, 14.29) | 0.018 | 3.66 (1.08, 12.42) | 0.037 |
| IV | 1 (4.5) | 1798.1 | 55.6 (7.8, 394.8) | 3.69 (0.47, 28.65) | 0.213 | 2.17 (0.24, 19.26) | 0.487 |
| **Baseline ART–NRTI backbone** | | | | | | | |
| AZT | 13 (59.1) | 44894.4 | 29.0 (16.8, 49.9) | Ref. | | Ref. | |
| ABC | 4 (18.2) | 15777.7 | 25.4 (9.5, 67.5) | 0.94 (0.30, 2.88) | 0.908 | 0.72 (0.22, 2.42) | 0.599 |
| TDF | 5 (22.7) | 5219.7 | 9.5 (3.9, 22.8) | 0.27 (0.10, 0.79) | 0.016 | 0.34 (0.09, 1.23) | 0.099 |
| D4T | 0 (0.0) | 2968.8 | 0.0 | 0.0 | - - - | 0.0 | - - - |
| **IPT outcome** | | | | | | | |
| Completed | 18 (81.8) | 108200.9 | 16.6 (10.5, 26.4) | Ref. | | Ref. | |
| Stopped | 2 (9.1) | 999.0 | 200.2 (50.1, 800.5) | 14.54 (3.05, 69.13) | 0.001 | 25.96 (4.12, 163.48) | 0.001 |
| LTFU | 2 (9.1 | 6982.8 | 28.6 (7.2, 114.5) | 1.26 (0.26, 6.16) | 0.773 | 1.36 (0.27, 6.91) | 0.707 |
| Died | 0 (0.0) | 178.0 | 0.0 | 0.0 | - - - | 0.0 | |
| **Baseline CD4** | | | | | | | |
| <200 | 4 (18.2) | 26525.2 | 15.1 (5.7, 40.2) | Ref. | | | |
| 200–350 | 3 (13.6) | 20573.8 | 14.6 (4.7, 45.2) | 1.01 (0.23, 4.52) | 0.990 | | |
| 350–500 | 2 (9.1) | 12326.9 | 16.2 (4.1, 64.9) | 1.06 (0.19, 5.82) | 0.943 | | |
| 500+ | 7 (31.8) | 21152.6 | 33.1 (15.8, 69.4) | 2.19 (0.63, 7.58) | 0.217 | | |
| Not done | 6 (27.3) | 35782.1 | 16.8 (7.5, 37.3) | 1.27 (0.33, 4.85) | 0.730 | | |
| **Duration on ART at IPT initiation** | | | | | | | |
| <6 months | 2 (9.1) | 10817.5 | 18.5 (4.6, 73.9) | Ref. | | | |
| 6-<12 months | 1 (4.5) | 7753.1 | 12.9 (1.8, 91.6) | 0.73 (0.08, 8.05) | 0.796 | | |
| 12-<18 months | 2 (9.1) | 4537.4 | 44.1 (11.0, 176.2) | 2.32 (0.32, 16.51) | 0.402 | | |
| 18-<24 months | 0 (0.0) | 5574.2 | 0.0 | 0.0 | - - - | | |
| 24-<30 months | 1 (4.5) | 5567.0 | 18.0 (2.5, 127.5) | 1.01 (0.09, 11.16) | 0.994 | | |
| 30 -<36 months | 1 (4.5) | 9002.2 | 11.1 (1.6, 78.9) | 0.60 (0.05, 6.70) | 0.680 | | |
| 36+ months | 15 (68.2) | 73109.3 | 20.5 (12.4, 34.0) | 1.16 (0.26, 5.09) | 0.848 | | |
| **Baseline ART–other** | | | | | | | |
| Efavirenz | 12 (54.5) | 60929.1 | 19.7 (11.2, 34.7) | Ref. | | | |
| Nevirapine | 9 (40.9) | 51457.8 | 17.5 (9.1, 33.6) | 0.92 (0.38, 2.22) | 0.844 | | |

*(Continued)*

**Table 3.** (Continued)

|  | TB diagnosis n (%) | Total person months | Crude incidence rate per 100,000 person months (95% CI) | Crude HR* (95% CI) | p-value | Adjusted HR* (95% CI) | p-value |
|---|---|---|---|---|---|---|---|
| PI | 1 (4.5) | 3435.4 | 29.1 (4.1, 206.6) | 1.68 (0.22, 13.01) | 0.618 |  |  |
| DTG | 0 (0.0) | 538.4 | 0.0 | 0.0 | --- |  |  |

The variables that were included and retained in the multivariable Frailty model were age group, baseline WHO stage, baseline ART regimen and IPT outcome. After adjusting for these factors, the incidence of TB did not significantly differ between age groups and ART regimen types. However, the incidence remained higher among patients in WHO stage III compared to those in stage II (adjusted hazard ratio (aHR) (95% CI): 3.66 (1.08, 12.42), p = 0.037); and higher among those who discontinued IPT compared to those who completed (aHR (95% CI): 25.96 (4.12, 163.48), p = 0.001).

## Discussion

In this study, we followed up 2634 PLHIV on ART who initiated IPT for 116360.7 person-months. A 92.8% IPT completion rate was achieved and a cumulative incidence of 0.8%, an incidence rate of 18.9 per 100,000 person months of TB among PLHIV on ART who initiated IPT was recorded. TB incidence rate after IPT initiation was higher among males than females despite a higher number of TB cases among females in the study. The median time to TB incidence from IPT initiation date was 18.5 months.

The overall TB incidence in this study (equivalent to 0.019 per 100 person years) is lower than those reported by most studies. For example, studies in Zimbabwe (1.06 per 100 person years), South Africa (2.3 per 100 person years), Botswana (0.8 cases per 100 person years) and Indonesia (1.09 per 100 person years) all had higher TB incidences [5, 27, 31, 32] among PLHIV who initiated IPT. TB incidence in this study was also lower than a similar study in Ethiopia (n = 2524, 0.21/100 person years) and another in Tanzania (n = 68,378, 2.7/100 person years); both of which reported incident TB among PLHIV on ART and IPT exposed [21, 22]. Importantly, the Ethiopian study included both pre-and post-ART PLHIV and the Tanzanian study, both IPT exposed and non-exposed PLHIV, which could have contributed to a higher TB incidence. In contrast, the TB incidence in this study was higher than the findings by Nyathi et al. in Zimbabwe [5] with 0 cases recorded following IPT initiation among 214 PLHIV on ART. As Geremew et al. argue, these differences likely result from individual variances among PLIHV on ART such as adherence levels, socioeconomic gradients and country specific TB endemicity [4]. Nonetheless, our findings suggest that TB incidence remains low three years after initiation of IPT among PLHIV on ART.

The low TB incidence in this study is likely due to good adherence to both ART and IPT. Indeed, the IPT completion rate recorded was high, at 92.8%. TASO COEs are special HIV clinics that integrate multi-modal adherence support that could have contributed to high IPT completion rates. PLHIV on ART who initiated but discontinued IPT had higher hazards of incident TB compared to those who completed (see Figs 1 & 2), further underscoring the need to ensure IPT completion. As Ishani et al., and Salazar et al. argue, it is important that individuals who are initiated on IPT are supported to complete the course for maximal benefit. It is perhaps unsurprising that with a completion rate of 94%, the Zimbabwean study recorded zero cases among PLHIV who initiated IPT [5].

The rate of IPT non-completion (7.2%) in the study is comparable to that reported by Sensalire et al. from a program setting [33], but higher than that reported by Lwevola in Eastern

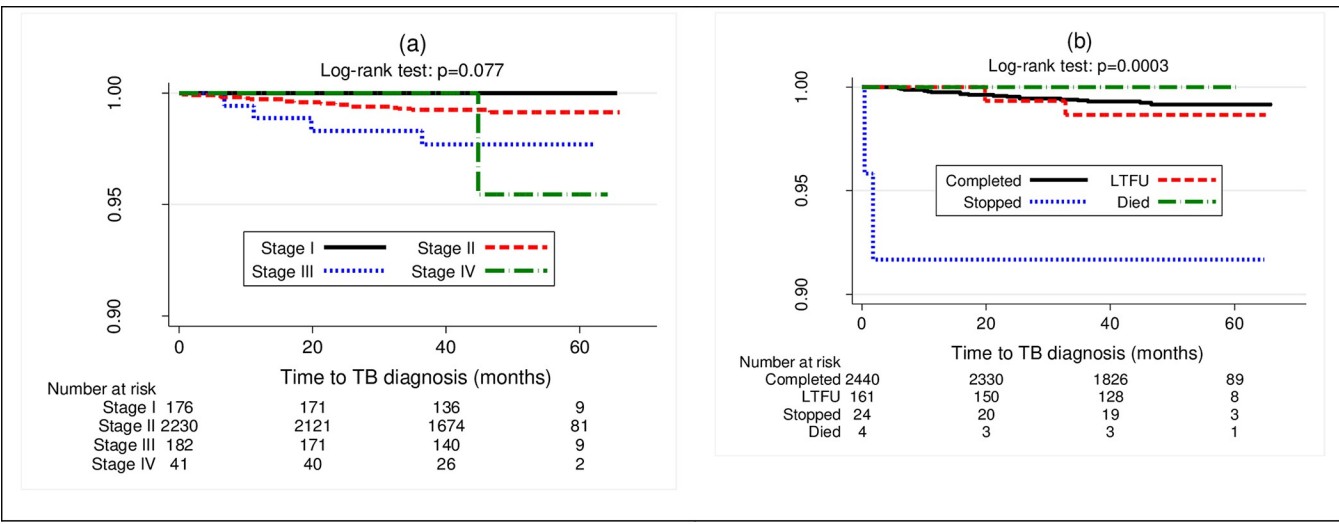

**Fig 1.** Kaplan Meier curves showing proportion surviving from TB infection by WHO HIV clinical stage (panel a) and IPT outcomes (panel b).

Uganda [34] and lower (42%) than that reported by Kalema et al. [35]. Among the patients who discontinued IPT, 20% stopped because of active TB implying failure to exclude TB disease prior to IPT initiation. This underlines one of the challenges to IPT scale up among PLHIV, the inability to exclude active disease, and further validates the need for improved TB screening tools and active patient surveillance after IPT initiation [9, 10, 23]. Other reasons for IPT non-completion such as drug stockouts are the focus of national TB quality improvement programs [24].

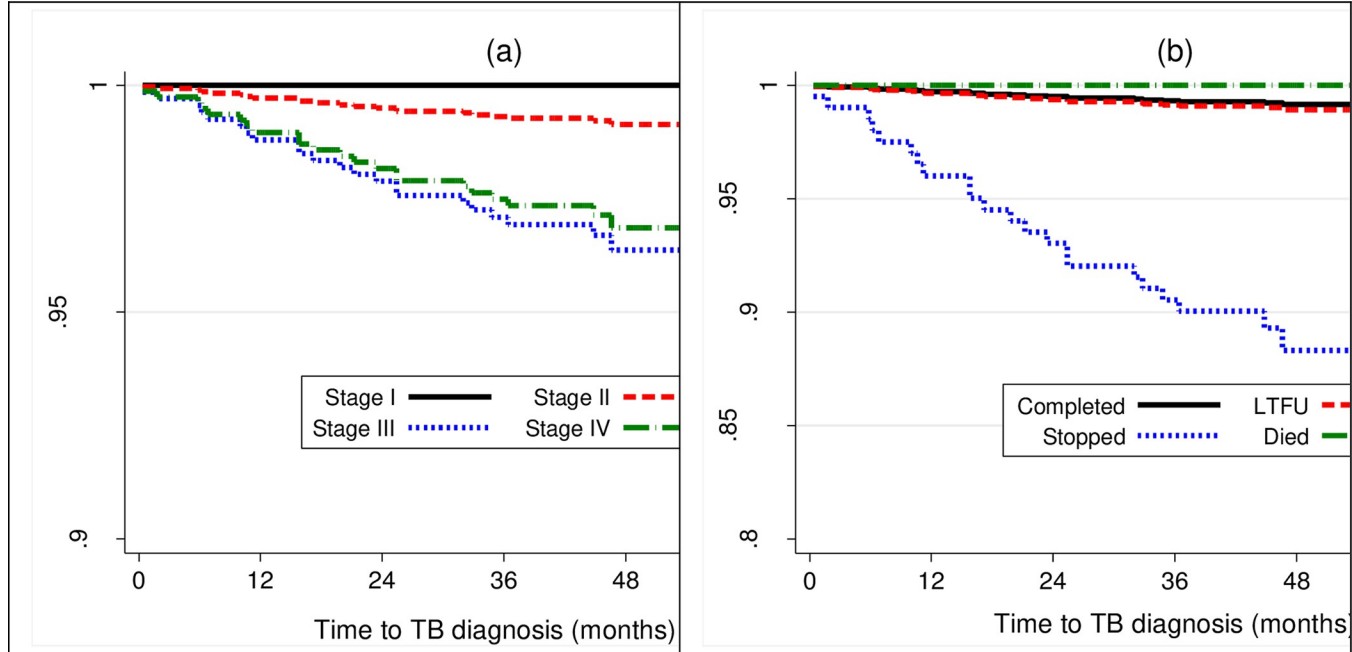

**Fig 2.** Shows probability of surviving TB disease, comparing WHO stages (panel a) and IPT outcomes (panel b) adjusted for clustering and other factors.

Another important finding from this study was that individuals in WHO HIV clinical stage III were more susceptible to active TB disease development compared to those in stage II (see Figs 1 & 2). Comparatively, both the Tanzanian and Ethiopian studies reported association between WHO HIV clinical stage III and active TB disease development [21, 22]. Similarly, Umeokonkwo et al. also reported an association between TB incidence in their study with WHO clinical stages III and IV [36]. WHO clinical stages III and IV are indicators of advanced HIV disease and likely a compromised immune status [37]. As such, individuals who present in these stages often possess cell of differentiation (CD4) count of less than 200 cells/μL, making them substantially vulnerable to developing TB disease [38] regardless of IPT status. Perhaps, findings by Golub et al. substantiate this suggestion [39]. They reported an increased risk of TB incidence among PLHIV whose CD4 counts fell below 200 cells. Similarly, Samandari et al. also reported increased risk of TB incidence when CD4 was below 200 cells/μL [31]. Indeed, Sabasaba et al. reported that CD4 counts above 200 cells/μL were associated with reduced risk of active TB disease development [21]. Adhering to both ART and IPT with fidelity likely increases the CD4 count, as Abossie et al. [7] and Semu et al. [22] reported in their studies. This in turn, restores the integrity of the immune system, hence protection against opportunistic infections such as TB. The individuals who developed active TB disease in our study, likely experienced sub-optimal adherence to both IPT and ART, as evidenced by higher number of cases among those who discontinued the therapy. However, it is important to note that we were unable to rule out additional pathway which includes exposure to fresh bacilli [4] among the TB cases, due to insufficient data.

Finally, the time to TB disease diagnosis was averagely 18.9 months from IPT initiation, a similar finding to that by Semu et al. in Ethiopia [22]. It suggests the effectiveness of IPT in the first six to twelve months of therapy regardless of completion status, among PLHIV on ART.

There were three major study strengths. Firstly, the wide geographical distribution of TASO Centers across Uganda, implying that data are nationally representative. Secondly, the use of routine program data for this study provides evidence that likely reflects actual reality in the field. Finally, we also recognize that routine screening of individuals who received IPT during the follow-up visits, enabled identification of TB cases.

We also acknowledge some key study limitations such as: we used secondary and routine program data, inherently vulnerable to missing variables, leading to exclusion of certain participants who could have added more valuable data. In addition, data on TB exposure factors such as contact with chronic cough cases, smoking and occupation were not collected as these are not routinely documented. Furthermore, the IPT and ART care outcomes of individuals who were lost to follow up, missed appointment or were transferred out could not be ascertained and this could lead to inaccurate estimation of TB incidence in this population. Also, since IPT completion is not objectively measured routinely, we did not ascertain adherence to IPT; but relied on patient chart records of IPT completion that are drawn from patient self-reports, prescription and dispensing records. Furthermore, in six of the TB cases, the diagnosis was made clinically as per MOH Guidelines. Considering the complexity of TB diagnosis in PLHIV, alternative diagnoses could not be excluded and this could have potentially led to an over-estimation of TB incidence.

## Recommendations and conclusions

Despite the study limitations discussed previously, our study found a significantly lower TB incidence among PLHIV initiated on IPT compared to earlier studies after three years of follow-up. This is partly attributed to the high completion rates of IPT among our study participants. We therefore, recommend that HIV programs need to support all PLHIV initiating IPT

to complete the therapy. In addition, while our study provides nationally representative findings, we only included participants from TASO COEs, that provided additional adherence support which may not necessarily be available in public health facilities. Consequently, findings may be different in those settings.

## Supporting information

**S1 Dataset.**
(XLSX)

## Acknowledgments

The research team sincerely appreciates the tireless efforts of the following individuals, who ensured smooth implementation of the study through resource mobilization, training of research assistants, and supervision of data collection: Gordon Karukoma, Edward Ssimbwa, Esele Brian, Eunic Ajambo, Ogwang Calvin, Ronald Achidri, Topher Ogwang, Caren Mutalwa, Emma Buxton Maedero, Shamim Namukose, Bernard Ouma Namulanda, Shadrack Ekwaro, Among Marion Gladys, Ritah Kwagala, Ronald Musisi, Banura Rehema, Brumno Kanyonyozi, Daren Mutegeki, Asaba Amos Kugonza, Dickson Niwasasira, Polly Niwamanya, Emmanuel Twesigye, Patience Kembale, Kenneth Nyeko, Emmanuel Welikhe, Nassuuna Maria Rita, Moses Okech, Christine Navvuga, Sunday Clay, and Asuman Ssewaya. Finally, we acknowledge PEPFAR through CDC/USAID for their continued support toward TASO COE programs.

## Author Contributions

**Conceptualization:** Andrew Kazibwe, Bonniface Oryokot, David Kagimu, Yunus Miya, Irene Andia Biraro.

**Data curation:** Andrew Kazibwe, Levicatus Mugenyi, Abraham Ignatius Oluka.

**Formal analysis:** Andrew Kazibwe, Levicatus Mugenyi.

**Investigation:** Andrew Kazibwe, Bonniface Oryokot, Darlius Kato, Edmund Tayebwakushaba, Charles Odoi, Kizito Kakumba, Ronald Opito, Ceasar Godfrey Mafabi, Michael Ochwo, Robert Nkabala, Wilber Tusiimire, Agnes Kateeba Tusiime, Sarah Barbara Alinga, Yunus Miya, Michael Bernard Etukoit.

**Methodology:** Andrew Kazibwe, Bonniface Oryokot, Levicatus Mugenyi, Abraham Ignatius Oluka, Simple Ouma, Edmund Tayebwakushaba, Charles Odoi, Ronald Opito.

**Project administration:** Bonniface Oryokot.

**Resources:** Andrew Kazibwe.

**Supervision:** Andrew Kazibwe, Bonniface Oryokot, Darlius Kato, Edmund Tayebwakushaba, Charles Odoi, Kizito Kakumba, Ronald Opito, Ceasar Godfrey Mafabi, Michael Ochwo, Robert Nkabala, Wilber Tusiimire, Agnes Kateeba Tusiime, Sarah Barbara Alinga, Yunus Miya, Michael Bernard Etukoit.

**Validation:** Andrew Kazibwe, Bonniface Oryokot, Levicatus Mugenyi.

**Visualization:** Levicatus Mugenyi.

**Writing – original draft:** Andrew Kazibwe, Bonniface Oryokot, Levicatus Mugenyi, Simple Ouma, Irene Andia Biraro, Bruce Kirenga.

**Writing – review & editing:** Andrew Kazibwe, Bonniface Oryokot, Levicatus Mugenyi, Simple Ouma, Yunus Miya, Michael Bernard Etukoit, Irene Andia Biraro, Bruce Kirenga.

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
