## [Decision Letter · Decision Letter 0]

15 Dec 2021

PONE-D-21-33906Incidence of Tuberculosis among PLHIV on Antiretroviral therapy who initiated TB preventive therapy: a multi-center retrospective cohort studyPLOS ONE

Dear Dr. Oryokot ,

Thank you for submitting your manuscript to PLOS ONE. After careful consideration, we feel that it has merit but does not fully meet PLOS ONE’s publication criteria as it currently stands. Therefore, we invite you to submit a revised version of the manuscript that addresses the points raised during the review process.

We look forward to receiving your revised manuscript.

Kind regards,

Kevin Schwartzman

Academic Editor

PLOS ONE

Journal Requirements:

(TASO acknowledges the PEPFAR funding of the eleven TASO centers through CDC, USAID and respective prime partners. The research team also sincerely appreciates the tireless efforts of the following individuals who ensured smooth implementation of the study through resource mobilization, training of research assistants, and supervision of data collection: Gordon Karukoma, Edward Ssimbwa, Esele Brian, Eunic Ajambo, Ogwang Calvin,  Ronald Achidri, Topher Ogwang, Caren Mutalwa, Emma Buxton Maedero, Shamim Namukose, Bernard Ouma Namulanda, Shadrack Ekwaro, Among Marion Gladys, Ritah Kwagala, Ronald Musisi, Banura Rehema, Brumno Kanyonyozi, Daren Mutegeki, Asaba Amos Kugonza, Dickson Niwasasira, Polly Niwamanya, Emmanuel Twesigye, Patience Kembale, Kenneth Nyeko, Emmanuel Welikhe, Nassuuna Maria Rita, Moses Okech, Christine Navvuga, Sunday Clay, and Asuman Ssewaya..)

(The authors received no specific funding for this work)

a. If there are ethical or legal restrictions on sharing a de-identified data set, please explain them in detail (e.g., data contain potentially sensitive information, data are owned by a third-party organization, etc.) and who has imposed them (e.g., an ethics committee). Please also provide contact information for a data access committee, ethics committee, or other institutional body to which data requests may be sent.

b. If there are no restrictions, please upload the minimal anonymized data set necessary to replicate your study findings as either Supporting Information files or to a stable, public repository and provide us with the relevant URLs, DOIs, or accession numbers. For a list of acceptable repositories, please see http://journals.plos.org/plosone/s/data-availability#loc-recommended-repositories.

6. Please include a separate caption for each figure in your manuscript.

Additional Editor Comments :

Thank you for submitting this manuscript to PLOS ONE. It contains important and useful information for readers. However, the reviewers have made a number of suggestions for improvement; please address these carefully. The manuscript could also be shortened somewhat. Reviewer 2, in particular, has made suggestions for some unnecessary details or other elements that could be shortened or removed.

Reviewers' comments:

Reviewer's Responses to Questions

**Comments to the Author**

1. Is the manuscript technically sound, and do the data support the conclusions?

Reviewer #1: Yes

Reviewer #2: Partly

2. Has the statistical analysis been performed appropriately and rigorously? 

Reviewer #1: No

Reviewer #2: Yes

3. Have the authors made all data underlying the findings in their manuscript fully available?

Reviewer #1: No

Reviewer #2: Yes

4. Is the manuscript presented in an intelligible fashion and written in standard English?

Reviewer #1: Yes

Reviewer #2: No

5. Review Comments to the Author

Reviewer #1: Dear all

Thank you for involving me in the review of the intutile “ Incidence of Tuberculosis among PLHIV on Antiretroviral therapy who initiated TB preventive therapy: a multi-center retrospective cohort study” article which I found very interesting and which addressed an essential point of TPT which is the occurrence of TB after TPT in people living with HIV.

Major points

1-Introduction

Clarification of the national guideline in terms of screening of PLWHIV before initiation of TPT. Was TB ruled out clinically, or with a GeneXpert or with an X-ray?

2-Methods:

The usual cut-off point for multivariate regression is 0.20 or 0.25 and not 0.05

3- Results

The age variable: the reference age under 15 years old could be reviewed because in PHIV the child under 15 years old is naturally more vulnerable than an adult subject

Minor revisions

1- There are some unannounced abbreviations

2- The writing of MOH or MoH?

3- Specify the type of multivariate analysis: logistic regression or mixed regression

Ideally the mixed method is the one recommended to take into account the herogeneity between the centers. But failing that, a logistic regression can be carried out if we consider that the sites are almost identical

References

Some suggested references that could be added

1- Economic and modeling evidence for tuberculosis preventive therapy among people living with HIV: A systematic review and meta-analysis Aashna Uppal1,2,3, Samiha RahmanID 1,2,3, Jonathon R. CampbellID 1,2,3, Olivia OxladeID3,Dick MenziesID1,2,3*

2- The latent tuberculosis cascade-of-care among people living with HIV: A systematic review and meta-analysis. Mayara Lisboa Bastos, Luca Melnychuk, Jonathon R. Campbell, Olivia Oxlade, Dick Menzies

PLoS Med. 2021 Sep; 18(9): e1003703. Published online 2021 Sep 7. doi: 0.1371/journal.pmed.1003703

Reviewer #2: Incidence of Tuberculosis among PLHIV on Antiretroviral therapy who initiated TB

preventive therapy: a multi-center retrospective cohort study

This is a cohort study on medium to long-term (36 months minimum) follow up of PLH under ART having had complete or partial TPT. Data derive from routinely collected information under programmatic conditions. This is a relevant subject, as the duration of protection from TPT is controversial, with studies in various settings having shown conflicting results. Thus, there is no consensus on the need for long-term, repeated or even lifelong use of TPT in PLH.

The analysis and conclusions are sound. The treatment completion rate under routine condition was very high (>92%) and the incidence rate was low among those who completed treatment (and overall, as most did complete) over time. CD4, viral load and duration/type of ART had no effect, although clinical stage of disease had. Limitations (mainly possible bias from exclusions and absence of information on reexposure to TB index-patients) and strengths of the study are well discussed. However, the manuscript would benefit from some rewriting. General suggestions for this follow, but I would advise careful language review.

Title (and throughout the text): consider using the term isoniazide preventive therapy (or treatment) as this is what was analyzed, other TPT were not assessed. Also consider using the TBI instead of LTBI, as now suggested by WHO.

Abstract: Ugandan setting does not need to be discussed in the background of the abstract, as this is a matter of interest around the globe. Adjusted (instead of crude) HR could be displayed here. The conclusion in the abstract is misleading. The authors could highlight here the main messages: incidence rates of TB were low overtime after one course of IPT, and this was mainly due to high completion rates.

The introduction could be substantially shortened. Firstly, the need for TPT in PLH and other high-risk group is well established, and the authors do not need to make a strong case on this, just a couple of sentences are sufficient. They could cite the literature on the specific subject of their analysis (for example, Samandari et al https://doi.org/10.1097/QAD.0000000000000535, Golub et al https://doi.org/10.1093/cid/ciu849) and emphasize the need for more long-term follow up studies to generate evidence on the ideal duration of TPT. Additionally, this is a relevant matter outside Uganda as well. Thus, I recommend that the authors move some information on the Uganda national recommendations to the Methods section, under “Setting”. The Uganda case is an example of the interest of this subject, which is relevant around the globe. Finally, the Introduction needs updating (WHO report 2020 available).

In the methods section, please clarify if patients starting or using IPT in the study period were included. Later in the text it seems that this is those starting IPT. Please define “stop therapy”. The heading “sample size” is not necessary, the information contained there should be moved to the Results section. Table 1 could be moved to supplement material. The subheading “patient enrolment” could be renamed data extraction. Participants were not exactly enrolled, this is a retrospective study on existing routine data. The exposure variables should be defined in the methods section, after definition of outcomes. It is not clear how the authors “triangulated” data with de-identified registries. Variables cited in the methods section that are not reported in the results section should be suppressed from there (ART status, type of TB, microbiological tests).

Please clarify the following:

• Are registered individuals those who were prescribe or those who initiated IPT? Meaning, are those who did not start (primary drop-out) included?

• Line 47: How can death be based on clinical encounter?

• Line 123: “All PLHIV who received TPT from 1st January 2016 to 30th June 2018”… This is a 30-month period but then you refer to 3 years or 36 months. I believe this means that the period of study is those starting IPT, and then they were followed up until 2021, for a minimum of 3 years, as very clearly explained in the abstract (please use same wording).

• Was treatment interruption considered at any number of doses taken? Early studies by Comstock suggest that 80% of doses are protective. Do the authors have information on the mean number of doses taken by those who did not complete IPT?

• Do the authors have information on cause of death? How was death information collected? I suggest a sensitivity analysis including deaths as a TB outcome (instead of censoring). TB is still the most frequent cause of death among PLH.

In the Results section, please inform overall incidence rate (per 100k person-month) as defined in the methods section and shown in the abstract, first paragraph of discussion and table 3. This information is more meaningful than the cumulative incidence. The p values are unnecessary, as the CI are displayed. Please consider a “survival curve” figure to illustrate TB cases (I would choose those completing versus those not completing IPT, the difference is impressive, as expected).

In table 3, do those lost to follow up include all censored? Please add footnote to tables, with abbreviations and this kind of clarification.

The discussion needs to be substantially shortened. The main message seems to be that completed TPT (or preferably, IPT) is associated with sustained protection (at least in the period of the study), as TB incidence rates remain low after a minimum period of 3 years. Maybe also that the mean time to TB development is 18 months. Also worthy discussing effect of clinical indicators of severity of HIV disease and absence of effect of other indicators of severity of disease, such as CD4 or type/duration of ART. None of these latter findings is surprising or novel, but the information that incident TB disease remains low after this long period is relevant and adds some light to the knowledge gap defined in the Introduction. This should be the main focus of the discussion. Do not include a detailed discussion (or even less speculation on why – this was not what you assessed) on every variable that was not associated with the outcome of interest. Above all, do not repeat detailed findings in the Discussion section. You can just summarize the main results in the first paragraph. For the sake of clarity for the reader, the authors could calculate the corresponding incidence rates in Tanzania and Ethiopia in the same unit as their findings (cases per 100,000 persons-months). A striking finding that also needs discussion, and explains the low incidence rate overall of TB, is the extremely high treatment completion rates. Mechanisms of drug action and BCG vaccination were not assessed and the discussion on these aspects is not useful. This is unusual under routine conditions. The support detailed in the methods section (lines 118-120) is probably related to these high completion rates and should be discussed. Limitations and strengths of the study should be incorporated to the discussion. An additional strength of the study is the active screening for incident TB in follow up visits.

Finally, why do the authors conclude that 3 years would be a potential timing for retreatment, if they found 18m as the median time to TB diagnosis? As the authors discuss, they do not have data on re-exposure, and it is not reasonable to speculate, based on their results, a potential timing for retreatment. I also disagree that the authors identified populations who might benefit from retreatment, this was not evaluated. Conclusions should be restricted to what was evaluated, anything else can be discussed, speculated, but not concluded.

Minor comments:

Please verify all abbreviations. TASO, cHR, WHO are used in abstract without definition. In the main text, other non-defined abbreviations appear, such as ART and aHR, please check all.

Consider changing “stopping” to discontinuation of treatment.

Also in the abstract, no space between numbers and th for dates. (1st, not 1 st)

I suggest changing patients enrolled to patients initiated on IPT.

Line 126 – should be in results (exclusion)

Many words with unjustified capital letters, such as the title and the variables in the Methods section.

Line 147 follow up

The discussion states “In the present study” at least 5 times. This is not necessary. The reader will know you are discussing your results.

L. 318 – discussed highlighted

6. PLOS authors have the option to publish the peer review history of their article (what does this mean?). If published, this will include your full peer review and any attached files.

Reviewer #1: No

Reviewer #2: No

---

## [Author Response · Author response to Decision Letter 0]

28 Jan 2022

1. The funding information was removed from the manuscript

2. Ethical statement only appears once, under the methodology section

3. Manuscript has been aligned with the journal standard

---

## [Decision Letter · Decision Letter 1]

7 Mar 2022

PONE-D-21-33906R1Incidence of Tuberculosis among PLHIV on Antiretroviral therapy who initiated isoniazid preventive therapy: a multi-center retrospective cohort studyPLOS ONE

Dear Dr. Oryokot ,

Thank you for submitting your manuscript to PLOS ONE. After careful consideration, we feel that it has merit but does not fully meet PLOS ONE’s publication criteria as it currently stands. Therefore, we invite you to submit a revised version of the manuscript that addresses the points raised during the review process.

Thank you for submitting your revised manuscript, and for carefully addressing the previous reviewer comments and suggestions. There remain a few, relatively straightforward areas for improvement, as indicated by the reviewers. Please address these in what will likely be your final revisions.==============================

We look forward to receiving your revised manuscript.

Kind regards,

Kevin Schwartzman

Academic Editor

PLOS ONE

Journal Requirements:

Reviewers' comments:

Reviewer's Responses to Questions

**Comments to the Author**

1. If the authors have adequately addressed your comments raised in a previous round of review and you feel that this manuscript is now acceptable for publication, you may indicate that here to bypass the “Comments to the Author” section, enter your conflict of interest statement in the “Confidential to Editor” section, and submit your "Accept" recommendation.

Reviewer #1: All comments have been addressed

Reviewer #2: (No Response)

2. Is the manuscript technically sound, and do the data support the conclusions?

Reviewer #1: Yes

Reviewer #2: Yes

3. Has the statistical analysis been performed appropriately and rigorously? 

Reviewer #1: Yes

Reviewer #2: Yes

4. Have the authors made all data underlying the findings in their manuscript fully available?

Reviewer #1: Yes

Reviewer #2: Yes

5. Is the manuscript presented in an intelligible fashion and written in standard English?

Reviewer #1: Yes

Reviewer #2: Yes

6. Review Comments to the Author

Reviewer #1: it would be desirable to justify the choice of the age of less than 15 years as a reference in the multivariate analysis method

Reviewer #2: Thank you for addressing the issues raised by the reviewers. The revised version of the manuscript is very interesting.

I would recommend some very minor changes:

1. Incident TB has been attributed to new infections more than reactivation of old infections by many. Your data seem to show the contrary. Anyway, it would be interesting to cite this controversial subject in your discussion (one sentence) as you have data from a high transmission setting, as I understand.

2. Some capital letters remais (e.g., variables in the subheading)

3. MoH still appears at least in one sentence. Plaese revise carefully

4. that followed by another that in one sentence (lines 422 and 423) - please reword.

5. Reviewer 1 has recommended some publications. I couldn't see a reply to their suggestion, that you did not accept, apparently.

Congratulations on your work!

7. PLOS authors have the option to publish the peer review history of their article (what does this mean?). If published, this will include your full peer review and any attached files.

Reviewer #1: No

Reviewer #2: No

---

## [Author Response · Author response to Decision Letter 1]

10 Mar 2022

We have appropriately responded to the reviewers' concerns

---

## [Editor Report · Decision Letter 2]

18 Mar 2022

Incidence of Tuberculosis among PLHIV on Antiretroviral therapy who initiated isoniazid preventive therapy: a multi-center retrospective cohort study

PONE-D-21-33906R2

Dear Dr. Oryokot ,

We’re pleased to inform you that your manuscript has been judged scientifically suitable for publication and will be formally accepted for publication once it meets all outstanding technical requirements.

Kind regards,

Kevin Schwartzman

Academic Editor

PLOS ONE

Additional Editor Comments (optional):

Thank you for making these last revisions.
---

## [Editor Report · Acceptance letter]

4 May 2022

PONE-D-21-33906R2 

Incidence of Tuberculosis among PLHIV on Antiretroviral therapy who initiated isoniazid preventive therapy: a multi-center retrospective cohort study 

Dear Dr. Oryokot :

I'm pleased to inform you that your manuscript has been deemed suitable for publication in PLOS ONE. Congratulations! Your manuscript is now with our production department. 

Kind regards, 

on behalf of

Dr. Kevin Schwartzman 

Academic Editor

PLOS ONE